# SwipeRing: Gesture Typing on Smartwatches Using a Segmented QWERTY Around the Bezel

Gulnar Rakhmetulla*
Human-Computer Interaction Group
University of California, Merced

Ahmed Sabbir Arif†
Human-Computer Interaction Group
University of California, Merced

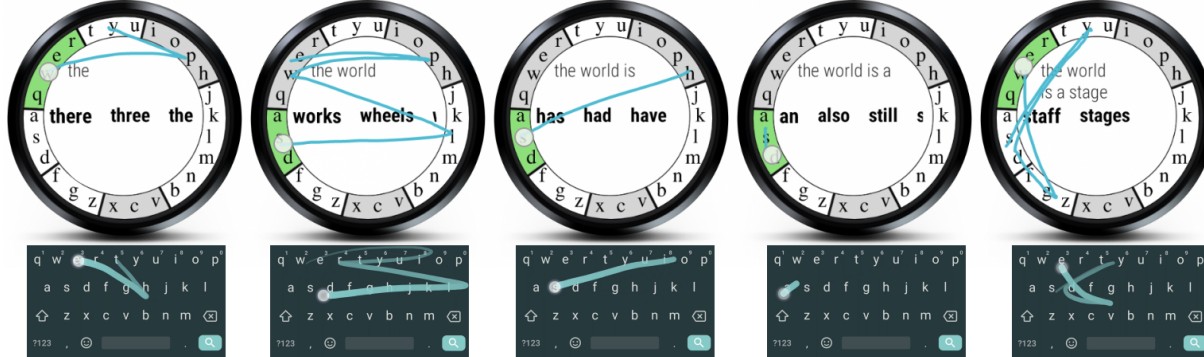

Figure 1: SwipeRing arranges the standard QWERTY layout around the edge of a smartwatch in seven zones. To enter a word, the user connects the zones containing the target letters by drawing gestures on the screen, like gesture typing on a virtual QWERTY. A statistical decoder interprets the input and enters the most probable word. A suggestion bar appears to display other possible words. The user could stroke right or left on the suggestion bar to see additional suggestions. Tapping on a suggestion replaces the last entered word. One-letter and out-of-vocabulary words are entered by repeated strokes from/to the zones containing the target letters, in which case the keyboard first enters the two one-letter words in the English language (see the second last image from the left), then the other letters in the sequence in which they appear in the zones (like multi-tap). Users could also repeatedly tap (instead of stroke) on the zones to enter the letters. The keyboard highlights the zones when the finger enters them and traces all finger movements. This figure illustrates the process of entering the phrase *"the world is a stage"* on the SwipeRing keyboard (upper sequence) and on a smartphone keyboard (bottom sequence). We can clearly see the resemblance of the gestures.

## ABSTRACT

Most text entry techniques for smartwatches require repeated taps to enter one word, occupy most of the screen, or use layouts that are difficult to learn. Users are usually reluctant to use these techniques since the skills acquired in learning cannot be transferred to other devices. SwipeRing is a novel keyboard that arranges the QWERTY layout around the bezel of a smartwatch divided into seven zones to enable gesture typing. These zones are optimized for usability and to maintain similarities between the gestures drawn on a smartwatch and a virtual QWERTY to facilitate skill transfer. Its ring-shaped layout keeps most of the screen available. We compared SwipeRing with C-QWERTY that uses a similar layout but does not divide the keys into zones or optimize for skill transfer and target selection. In a study, SwipeRing yielded a 33% faster entry speed (16.67 WPM) and a 56% lower error rate than C-QWERTY.

**Index Terms:** Human-centered computing—Text input; Human-centered computing—Gestural input

## 1 INTRODUCTION

Smartwatches are becoming increasingly popular among mobile users [32]. However, the absence of an efficient text entry technique for these devices limits smartwatch interaction to mostly controlling applications running on smartphones (e.g., pausing a song on a media player or rejecting a phone call), checking notifications on incoming text messages and social media posts, and using them as fitness trackers to record daily physical activity. Text entry on smartwatches is difficult due to several reasons. First, the smaller key sizes of miniature keyboards make it difficult to tap on the target keys (the "fat-finger problem" [55]), resulting in frequent input errors even when augmented with a predictive system. Correcting these errors is also difficult, and often results in additional errors. To address this, many existing keyboards use a multi-action approach to text entry, where the user performs multiple actions to enter one letter (e.g., multiple taps [15], chords [41]). This increases not only learning time but also physical and mental demands. Besides, most existing keyboards cover much of the smartwatch touchscreen (50–85%), reducing the real estate available to view or interact with the elements in the background. Many keyboards for smartwatches that use novel layouts [1] do not facilitate skill transfer. That is, the skills acquired in learning new keyboards are usually not usable on other devices. This discourages users from learning a new technique. Further, most of these keyboards were designed for square watch-faces, thus do not always work on round screens. Finally, some techniques rely on external hardware, which is impractical for wearable devices.

To address these issues, we present SwipeRing, a ring-shaped keyboard that sits around the smartwatch bezel to enable gesture typing with the support of a statistical decoder. Its ring-shaped layout keeps most of the touchscreen available to view additional information and perform other tasks. It uses a QWERTY-like layout divided into seven zones that are optimized to provide comfortable areas to initiate and release gestures, and to maintain similarities between the gestures drawn on a virtual QWERTY and SwipeRing to facilitate skill transfer.

*e-mail: grakhmetulla@ucmerced.edu
†e-mail: asarif@ucmerced.edu

The remainder of the paper is organized as follows. First, we discuss the motivation for the work, followed by a review of the literature in the area. We then introduce the new keyboard and discuss its rigorous optimization process. We present the results of a user study that compared the performance of the proposed keyboard with the C-QWERTY keyboard that uses a similar layout but does not divide the keys into zones or optimize for skill transfer and target selection. Finally, we conclude the paper with potential future extensions of the work.

## 2 MOTIVATION

The design of SwipeRing is motivated by the following considerations.

### 2.1 Free-up Touchscreen Real-Estate

On a 30.5 mm circular watch, a standard QWERTY layout without and with the suggestion bar occupy about 66% (480 mm$^2$) and 85% (621 mm$^2$) of the screen, respectively (Fig. 2). On the same device, our technique, SwipeRing, occupies only about 36% (254.34 mm$^2$) of the screen, almost half the QWERTY layout. Saving screen space is important since the extra space could be used to display additional information and to make sure that the interface is not cluttered, which affects performance [24]. For example, the extra space could be used to display the email users are responding to or more of what they have composed. The former can keep users aware of the context. The latter improves writing quality [58, 60] and performance [18]. Numerous studies have verified this in various settings, contexts, and devices [42–44, 46–49].

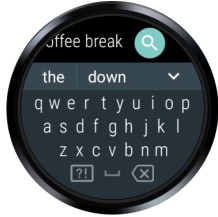 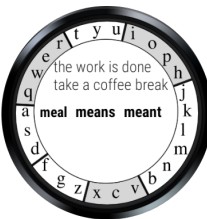

Figure 2: When entering the phrase "the work is done take a coffee break" with a smartwatch QWERTY, only the last 10 characters are visible (left), while the whole phrase is visible (37 characters) with SwipeRing (right). Besides, there is extra space available below the floating suggestion bar to display additional information.

### 2.2 Facilitate Skill Transfer

The skill acquired in using a new smartwatch keyboard is usually not transferable to other devices. This discourages users from learning a new technique. The keyboards that attempt to facilitate skill transfer are miniature versions of QWERTY that are difficult to use due to the small key sizes. To mitigate this, most of these keyboards rely on statistical decoders, making the entry of out-of-vocabulary words difficult, or impossible. SwipeRing uses a different approach. Although gesture typing is much faster than tapping [30], it is not a dominant text entry method on mobile devices. SwipeRing strategically arranges the letters in the zones to maintain gesture similarities between the gestures drawn on a virtual QWERTY and SwipeRing to enable performing the same (or very similar) gesture to enter the same word on various devices. The idea is that it will encourage gesture typing by facilitating skill transfer from smartphones to smartwatches and vice versa.

### 2.3 Increase Usability

As a result of an optimization process, the layout is strategically divided into seven zones, accounting for the mean contact area in index

finger touch (between 28.5 and 33.5 mm$^2$ [56]), to facilitate comfortable and precise target selection during text entry [38]. The zones range between 29.34 and 58.68 mm$^2$ (lengths between 9.0 and 18.0 mm), which are within the recommended range for target selection on both smartphones [26, 31, 39] and smartwatches (7.0 mm) [10]. SwipeRing employs the whole screen for drawing gestures, which is more comfortable than drawing gestures on a miniature QWERTY. Unlike most virtual keyboards, SwipeRing requires users to slide their fingers from/to the zones instead of tapping, which also makes target selection easier. Existing work on eyes-free bezel-initiated swipe for the ring-shaped layouts revealed that the most accurate layouts have 6–8 segments [61]. SwipeRing enables the entry of out-of-vocabulary words through a multi-tap like an approach [9], where users repeatedly slide their fingers from/to the zone that contains the target letter until the letter is entered (see Section 5.2 for further details). Besides, research showed that radial interfaces on circular devices visually appear to take less space even when they occupy the same area as rectangular interfaces, which not only increases clarity but also makes the interface more pleasant and attractive [45].

### 2.4 Face Agnostic

Since SwipeRing arranges the keys around the edge of a smartwatch, it works on both round and square/rectangular smartwatches. To validate this, we investigated whether the gestures drawn on a square smartwatch and a circular smartwatch are comparable to the ones drawn on a virtual QWERTY. A Procrustes analysis on the 10,000 most frequent words drawn on these devices, SwipeRing yielded a score of 114.81 with the square smartwatch and 118.48 with the circular smartwatch. This suggests that the gestures drawn on these devices are very similar. In fact, the square smartwatch yielded a slightly better score than the circular smartwatch (the smaller the score, the better the similarity, Section 4.2), most likely because the shapes of these devices are similar (Fig. 3).

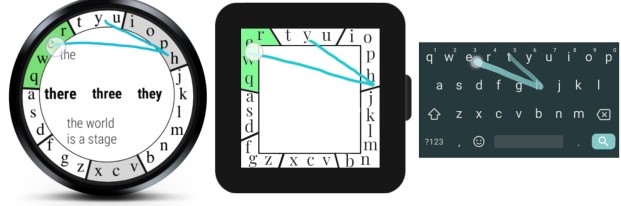

Figure 3: Gestures for the most common word "the" on a circular SwipeRing, a square SwipeRing, and a virtual QWERTY.

## 3 RELATED WORK

This section covers the most common text entry techniques for smartwatches. Tables 1 and 2 summarize the performance of some of these techniques. For a comprehensive review of existing text entry techniques for smartwatches, we refer to Arif et al. [1].

### 3.1 QWERTY Layout

Most text entry techniques for smartwatches are miniature versions of the standard QWERTY that use multi-step approaches to increase the target area. ZoomBoard [37] displays a miniature QWERTY that enables iterative zooming to enlarge regions of the keyboard for comfortable tapping. SplitBoard [19] displays half of a QWERTY keyboard so that the keys are large enough for tapping. Users flick left and right to see the other half of the keyboard. SwipeBoard [5] requires two swipes to enter one letter, the first to select the target letter region and the second towards the letter to enter it. DriftBoard [50] is a movable miniature QWERTY with a fixed cursor point. To enter text, users drag the keyboard to position the intended key

Table 1: Average entry speed (WPM) of popular keyboards for smartwatches from the literature (only the highest reported speed in the last block or session are presented, when applicable) along with the estimated percentage of touchscreen area they occupy.

| Method | Used Device | Screen Occupancy | Entry Speed (WPM) |
|---|---|---|---|
| Yi et al. [62] | Watch 1.56" | 50% | 33.6 |
| WatchWriter [13] | Watch 1.30" | 85% | 24.0 |
| DualKey [16] | Watch 1.65" | 80% | 21.6 |
| SwipeBoard [5] | Tablet | 45% | 19.6 |
| SplitBoard [19] | Watch 1.63" | 75% | 15.9 |
| ForceBoard [20] | Phone | 67% | 12.5 |
| ZoomBoard [37] | Tablet | 50% | 9.3 |

Table 2: Average entry speed (WPM) of popular ring-shaped keyboards for smartwatches from the literature (only the highest reported speed in the last block or session are presented, when applicable) along with the estimated percentage of touchscreen area they occupy.

| Method | Used Device | Screen Occupancy | Entry Speed (WPM) |
|---|---|---|---|
| WrisText [12] | Watch 1.4" | 42.99% | 15.2 |
| HiPad [22] | VR | 39.70% | 13.6 |
| COMPASS [63] | Watch 1.2" | 36.07% | 12.5 |
| BubbleFlick [52] | Watch 1.37" | 52.06% | 8.0 |
| C-QWERTY$_{gesture}$ [7] | Watch 1.39" | 43.16% | 7.7 |
| Cirrin [25] | PC | 60.57% | 6.4 |
| InclineType [17] | Watch 1.6" | 38.64% | 5.9 |

within the cursor point. Some miniature QWERTY keyboards use powerful statistical decoders to account for the"fat-finger problem" [55]. WatchWriter [13] appropriates a smartphone QWERTY for smartwatches. It supports both predictive tap and gesture typing. Yi et al. [62] uses a similar approach with even smaller keyboards (30 and 35 mm). VelociWatch [54] also uses a statistical decoder, but enables users to lock in particular letters of their input to disable potential auto-corrections. Some techniques use variants of the standard QWERTY layout. ForceBoard [20] maps QWERTY to a 3×5 grid by assigning two letters to each key. Applying different levels of force on the keys enters the respective letters. DualKey [16] uses a similar layout, but requires users to tap with different fingers to disambiguate the input. It uses external hardware to differentiate between the fingers. DiaQWERTY [28] uses diamond-shaped keys to fit QWERTY in a round smartwatch at 10:7 aspect ratio. Optimal-T9 [40] maps QWERTY to a 3×3 grid, then disambiguates input using a statistical decoder. These techniques, however, occupy a substantial area of the screen real-estate, require multiple actions to enter one letter, or use prediction models that make the entry of out-of-vocabulary words difficult.

### 3.2 Other Layouts

There are a few techniques that use different layouts. Dunlop et al. [10] and Komninos and Dunlop [29] map an alphabetical layout to six ambiguous keys, then uses a statistical decoder to disambiguate input. It enables contextual word suggestions and word completion. QLKP [19] (initially designed for smartphones [21]) maps a QWERTY-like layout to a 3×3 grid. Similar to multi-tap [9], users tap on a key repeatedly until they get the intended letter. These techniques also occupy a substantial area of touchscreen real-estate and require multiple actions to enter one letter.

### 3.3 Ring-Shaped Layouts

There are some ring-shaped keyboards available for smartwatches. InclineType [17] places an alphabetical layout around the edge of the devices. To enter a letter, users first select the letter by moving the wrist, then tap on the screen. COMPASS [63] also uses an alphabetical layout, but does not use touch interaction. To enter text, users rotate the bezel to place one of the three available cursors on the desired letter, then press a button on the side of the watch. WrisText [12] is a one-handed technique, with which users enter text by whirling the wrist of the hand towards six directions, each representing a key in a ring-shaped keyboard with the letters in alphabetical order. BubbleFlick [52] is a ring-shaped keyboard for Japanese text entry. It enables text entry through two actions. Users first touch a key, which partitions the ring-shaped area inside the layout into four radial areas for the four kana letters on the key. Users then stroke towards the intended letter to enter it. A commercial product, TouchOne Keyboard Wear [53] divides an alphabetical layout into eight zones to let users enter text using a T9-like [14]

approach. It enables the entry of out-of-vocabulary words using an approach similar to BubbleFlick. Go et al. [11] designed an eyes-free text entry technique that enables users to EdgeWrite [59] on a smartwatch with the support of auditory feedback. Most of these techniques use a sequence of actions or a statistical decoder to disambiguate the input. Besides, most of these techniques are standalone, hence the skills acquired in using these keyboards are usually not usable on other devices.

Cirrin [35] is a pen-based word-level text entry technique for PDAs. It uses a novel ring-shaped layout with dedicated keys for each letter. To enter a word, users pass their pen through the intended letters. This approach has also been used on other devices [25]. C-QWERTY is a similar technique [7], but differs in letter arrangement, which is based on the QWERTY layout. To enter a word with C-QWERTY, users either tap on the letters in the word individually or drag the finger over them in sequence. While there are some similarities between Cirrin, C-QWERTY, and SwipeRing, the approach employed in the latter technique is fundamentally different. First, SwipeRing divides the layout into seven zones, thus does not require precise selection of the letters, but much larger zones. Both Cirrin and C-QWERTY, in contrast, use individual keys for each letter, thus require a precise selection of the keys. Second, like gesture typing on a smartphone, SwipeRing does not require users to go over the same letter (or the letters on the same zone) repeatedly if they appear in a word multiple times in sequence (such as, "oo" in "book"). But both Cirrin and C-QWERTY require users to go over the same letter repeatedly in such cases by sliding the finger out of the keyboard then sliding back to the key. We found out users use this strategy also for entering letters that have the respective keys placed side-by-side, as they are difficult to select consecutively due to the smaller size. Finally, SwipeRing is optimized to maintain gesture similarities between SwipeRing and a virtual QWERTY to facilitate skill transfer between devices.

## 4 LAYOUT OPTIMIZATION

SwipeRing maps the standard QWERTY layout to a ring around the edge of a smartwatch (Fig. 1). It places the left and the right-hand keys of QWERTY [36] to the left and right sides of the layout, respectively. Likewise, the top, home, and bottom row keys of QWERTY are placed at the top, middle, and bottom parts of the layout, respectively. Each letter is positioned at multiple of $360°/26$, resulting in an angular step of $13.80°$, starting with the letter 'q' at $180°$. This design was adapted to maintain a likeness to QWERTY to exploit the widespread familiarity with the keyboard to facilitate learning [21, 40]. We then grouped the letters into zones to improve the usability of the keyboard by facilitating precise target selection, further discussed in Section 4.3. In practice, the letters can be grouped in numerous different ways, resulting in a set of possible layouts $L$. However, the purpose here was to identify a particular layout $l \in L$ that ensures that the gestures drawn on the layout $l$

are similar to the ones drawn on a virtual QWERTY. This requires searching for an optimal letter grouping that maximizes gesture similarity. We introduce the following notation to formally define the optimization procedure. Let $g_Q(w)$ be the gesture used to enter a word $w$ on the virtual QWERTY and $g_{\text{SwipeRing}}(w; l)$ be the gesture used to enter the word $w$ on the layout $l$ of SwipeRing. Instead of maximizing the similarity between the gestures, we can equivalently minimize the discrepancy between the gestures, which we measure using a function $\psi$. Then, our problem is to find the layout $l$ that minimizes the following loss function $\mathscr{L}$:

$$\min_{\text{layout } l \in L} \mathscr{L}(l) = \sum_{w \in W} p(w)\, \psi\Big(g_Q(w), g_{\text{SwipeRing}}(w; l)\Big). \quad (1)$$

Here, the dissimilarity between the gestures is weighted by the probability of the occurrence of the word to assure that the gestures for the most frequent words are the most similar. We made several modeling assumptions and simplifications to efficiently optimize this problem, which are discussed in the following sections.

## 4.1 Gesture Modelling

We model each gesture as a piece-wise linear curve connecting the letters on a virtual QWERTY or the zones on SwipeRing. Therefore, the gesture for a word composed of $n$ letters can be seen as a $2 \times n$ dimensional matrix (Fig. 4), where each column contains coordinates $(x, y)$ of the corresponding letter. To simulate the drawn gesture on a virtual QWERTY for the word $w$, denoted as $g_Q(w)$, we connect the centers of the corresponding keys of the default Android QWERTY on a Motorola $G^5$ smartphone (24 cm$^2$ keyboard area)[1], producing unique gestures for each word. With SwipeRing, however, we account for the fact that a word can have multiple gestures forming a set $G_{\text{SwipeRing}}(w; l)$. The zones containing 4–6 letters are wide enough to enable initiating a gesture either at the center, left, or right side of the zone (Fig. 5), resulting in multiple possibilities. Therefore, we set the gesture $g_{\text{SwipeRing}}(w; l)$ to be the one that has minimal difference when compared to the gesture drawn on the virtual QWERTY measured by discrepancy function $\psi$:

$$g_{\text{SwipeRing}}(w; l) := \operatorname*{arg\,min}_{g \in G_{\text{SwipeRing}}(w; l)} \psi(g_Q(w), g). \quad (2)$$

## 4.2 Discrepancy Function

For the discrepancy function $\psi(g_1, g_2)$ between gestures $g_1$ and $g_2$, our requirement is to have a rotation and scale agnostic measure that attains a value of 0 if and only if $g_2$ is a rotated and re-scaled version

[1]Since most virtual QWERTY layouts maintain comparable aspect ratios and the gestures only loosely connect the keys, the gestures on different sized phones, keyboards, keys are comparable when the recognizer is size agnostic. A Procrustes analysis of the gestures drawn on five different sized phones with different keyboards and key sizes yielded results between 6 and 19, suggesting they are almost identical.

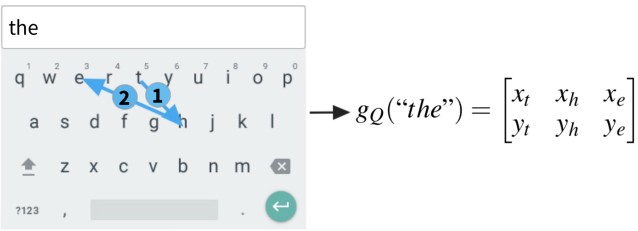

$$g_Q(\text{"the"}) = \begin{bmatrix} x_t & x_h & x_e \\ y_t & y_h & y_e \end{bmatrix}$$

Figure 4: The gesture for the most common word "the" on a virtual QWERTY and the respective $2 \times 3$ dimensional matrix.

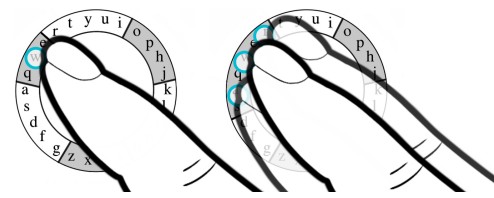

Figure 5: Gestures on the three letter zones are likely to be initiated from the center, while gesture on the wider zones (such as, a six letter zone) can be initiated from either the center or the two sides.

of $g_1$. One possible form of such $\psi$ function can be defined as yet another optimization problem of:

$$\psi(g_1, g_2) = \min_{R,\, \alpha} \| g_2 - \alpha R g_1 \|_F^2. \quad (3)$$

Where $\alpha$ and $R$ are the rescaling factor and the rotation matrix applied to a gesture, respectively, while $\|.\|_F$ is the Frobenius norm[2]. We recognize Equation 3 as an Ordinary Procrustes analysis problem, the solution of which is given in closed-form by Singular Value Decomposition [8]. Note that the value of $\psi(g_1, g_2)$ is within the range of $[0, \infty]$. Additionally, we restrict the rotation within the range of $\pm 45°$ since SwipeRing gestures that are rotated more than $45°$ in either direction are unlikely to look similar to their virtual QWERTY counterparts (Fig. 6).

| Best Match | Average Match | Worst Match |
|---|---|---|
| $\psi(g_1, g_2) = 0.86$ | $\psi(g_1, g_2) = 57.49$ | $\psi(g_1, g_2) = 133.53$ |

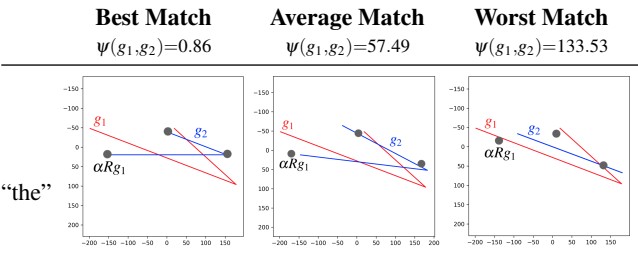

Figure 6: Gesture dissimilarity measures using the Procrustes loss function. The red curve represents the gesture for the word "the" on a virtual QWERTY ($g_1$), the blue curves represent gestures for the same word on a SwipeRing layout ($g_2$), and the gray dots show the optimal rotation and rescaling of the gesture ($g_1$) represented as ($\alpha R g_1$) to match ($g_2$).

## 4.3 Enumeration of All Possible Letter Groupings

The total number of possible letter groupings, and thus layouts, depends on how large we allow the groups to be. To determine this, we conducted a literature review of ambiguous keyboards that use linguistic models for decoding the input to find out whether the number of letters assigned per key or zone (the level of ambiguity) affects the performance of a keyboard. Table 3 displays an excerpt of our review, where one can see "somewhat" inverse relationship between the level of ambiguity and entry speed. Keyboards that assign fewer letters per key or zone yield a relatively better entry speed than those with more letters per key or zone. Based on this, we decided to assign 3–6 letters per zone. Although, this alone cannot determine the appropriate number of letters in each key since the performance of a keyboard depends on other factors, such as the layout and the reliability of the decoder, it gives a rough estimate.

[2]Frobenius norm is a generalization of Euclidean norm to the matrices, such as if $A$ is a matrix then $\|A\|_F^2 = \sum_{i,j} a_{ij}^2$

Table 3: Average entry speed of several ambiguous keyboards that map multiple letters to each key or zone.

| Method | Letters per Key | Entry Speed (WPM) |
|---|---|---|
| COMPASS [63] | 3 | 9–13 |
| HiPad [22] | 4–5 | 9.6–11 |
| WrisText [12] | 4–5 | 10 |
| Komninos, Dunlop [10, 29] | 3–6 | 8 |

Next, we discovered all possible shatterings of the ring-shaped string {qwertyuiophjklmnbvcxzgfdsa} into substrings of length 3–6 letters each, resulting in 4,355 different layouts in total, which constitute our search set $L$. Each possible shattering, such as {qwer}{tyu}{iophjk}{lmnbv}{cxzg}{fdsa}, represents one possible layout. We tested several of these layouts on a small smartwatch (9.3 cm$^2$ circular display) to investigate if the zones containing three letters are wide enough for precise target selection. Results showed that the zones range between 29.0 and 57.5 mm$^2$ (lengths between 9.0 and 18.0 mm), which are within the length recommended for target selection on both smartphones [26, 31, 39] and smartwatches (7.0 mm) [10]. Fig. 7 illustrate some of these layouts.

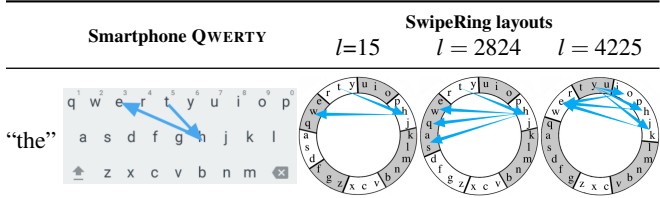

**Smartphone QWERTY**    **SwipeRing layouts**   $l=15$   $l=2824$   $l=4225$

"the"

Figure 7: Gesture typing the word "the" on a virtual QWERTY and three possible SwipeRing layouts. For the virtual QWERTY, the figure shows $g_Q$("the"). For the SwipeRing layouts, the figure shows all possible gestures for "the": $G_{SwipeRing}$("the", $l$). Notice how the gestures for the same word are different on different SwipeRing layouts.

## 4.4 Algorithm

To find the optimal layout, we simulated billions of gestures for the 10,000 most frequent words in the English language [57] on the 4,355 possible segmented SwipeRing layouts[3]. We then matched the gestures produced for each word on each layout with the gestures produced on a virtual QWERTY using the Procrustes analysis to pick the layout that yielded the best match score (118.48). The final layout (Fig. 1) scored, on average, 1.27 times better Procrustes value compared to the other possible layouts.

## 5 KEYBOARD FEATURES

This section describes some key features of the proposed keyboard.

## 5.1 Decoder

We developed a simple decoder to suggest corrections and display the most probable words in a suggestion bar. For this, we used a combination of a character-level language model and a word-level bigram model for the next word prediction. To this end, we calculate the conditional probability of the user typing the word $w$ given that

---

[3]We only used words that had more than one letter, there were 9,828 such words in the corpus.

---

**Algorithm 1:** Search for an optimal layout $l$.

**Input:** Possible grouping layouts $L = \{l_1, l_2, \dots\}$, word corpus $W = \{w_1, w_2, \dots\}$
**Function** OptLayout($L, W$):
   $\mathscr{L}_{min} \leftarrow \infty, l_{min} \leftarrow \infty$
   **for** layout $l \in L$ **do**
      $\mathscr{L} \leftarrow 0$
      **for** word $w \in W$ **do**
         $\mathscr{L} \leftarrow \mathscr{L} + p(w)\, \psi\Big(g_Q(w), g_{SwipeRing}(w; l)\Big)$
      **end**
      **if** $\mathscr{L} \leq \mathscr{L}_{min}$ **then**
         $\mathscr{L}_{min} \leftarrow \mathscr{L}, l_{min} \leftarrow l$
      **end**
   **end**

---

the previous word was $w_{n-1}$ and the current zone sequence is $s$:

$$P(w_n = w | s, w_{n-1}) = \frac{P(w_n = w, s, w_{n-1})}{P(s, w_{n-1})}$$
$$= \frac{\text{count}(w_n = w, w_{n-1}) \times \text{match}(M(w), s)}{\sum_{w'} \text{count}(w_n = w', w_{n-1}) \times \text{match}(M(w'), s))}. \quad (4)$$

Here, $M(w)$ is the sequence of zones that the user must gesture over to enter the word $w$ with SwipeRing, $\text{match}(s_1, s_2)$ is the indicator function that returns 1 if $s_2$ is a prefix of $s_1$ or 0 otherwise, and $\text{count}(w_n, w_{n-1})$ is the number of occurrences of a bigram $(w_n, w_{n-1})$ in the training corpus.

To predict the most probable word for a given zone sequence $s$ and previous word $w_{n-1}$, we compute $\arg\max_w P(w_n = w | s, w_{n-1})$ using the prefix tree (Trie) data structure. This implementation can output $k$ highest probable words, which we display in the suggestion bar. When no word has been typed yet, we use a unigram reduction of the model, otherwise, we use the bigram model trained on the COCA corpus [6]. Due to the limited memory capacity of the smartwatch, the Trie uses the 1,300 most probable bigrams: bigram models scale as the square of the number of words, thus quickly outrun the available memory on the device. If the Trie does not have a bigram containing the user's previous word $w_{n-1}$, we revert to the unigram predictions. Our language model is fairly simple, and more advanced models (involving neural nets, for instance) can be created. However, devising efficient language models for new keyboards is a research problem on its own and beyond the scope of our paper.

After obtaining the list of the most probable words, SwipeRing places up to 10 most probable words in the suggestion bar, automatically positioned in close proximity to the input area (Fig. 1). The suggestion bar automatically updates as the user continues gesturing. Once done, the most probable word from the list is entered. The user could select a different word from the suggestion bar by tapping on it, which replaces the last entered word. Although the user can only see 2–4 words in the suggestion bar due to the smaller screen, she can swipe left and right on the bar to see the remaining words.

## 5.2 One-Letter and Out-of-Vocabulary (OOV) Words

SwipeRing enables the entry of one-letter and out-of-vocabulary words through repeated taps or strokes from/to the zones containing the target letters. The keyboard first enters the two one-letter words in the English language "a" and "I", then the other letters in the sequence in which they appear in the zones, like multi-tap [9]. For instance, to enter the letter 'e', which is in the top-right zone containing the letters: 'q', 'w', 'e', and 'r' (Fig. 1), the user taps or slides the finger three times from the middle area to the zone or from the edge to the middle area (Fig. 8).

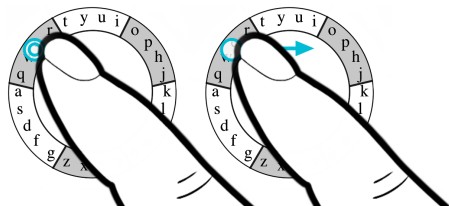

Figure 8: SwipeRing enables users to enter one-letter and out-of-vocabulary words by repeated strokes from/to the zones containing the target letters, like multi-tap (right). Users could also repeatedly tap on the zones (instead of strokes) to enter the letters (left).

### 5.3 Error Correction and Special Characters

SwipeRing automatically enters a *space* when a word is predicted or manually selected from the suggestion bar. During character-level text entry (to enter out-of-vocabulary words), users enter *space* by performing a right stroke inside the empty area of the keyboard. Tapping on the transcribed text deletes the last entry, either a word or a letter. The keyboard performs a *carriage return* or an *enter* action when the user double-taps on the screen. Currently, SwipeRing does not support uppercase letters, special symbols, numbers, and languages other than English. However, these could be easily added by enabling the user to long-press or dwell on the screen or the zones to switch back and forth between the cases and change the layout for digits and symbols. Note that the evaluation of novel text entry techniques without the support for numeric and special characters is common practice since it eliminates a potential confound [34].

## 6 USER STUDY

We conducted a user study to compare SwipeRing with C-QWERTY. C-QWERTY uses almost the same layout as SwipeRing but places 'g' at the NE corner, while SwipeRing places it at the SW corner (left side of the layout since 'g' on QWERTY is usually pressed with the left hand). Both layouts share the design goal of maintaining similarity to QWERTY by using the touch-typing metaphor of physical keyboards (keys assigned to different hands). This likely resulted in similar (but nonidentical) layouts. Studies showed that using a physical analogy/metaphor like this enables novices to learn a method faster by skill transfer [33, pp. 255–263]. Besides, C-QWERTY does not divide the keys into zones, optimize them for gesture typing and skill transfer, and uses a slightly different mechanism for gesture drawing approaches for the two are also different (Section 3.3). Hence, a comparison between the two will highlight the performance difference due to the contributions of this work.

### 6.1 Apparatus

We used an LG Watch Style smartwatch, 42.3×45.7×10.8 mm, 9.3 cm$^2$ circular display, 46 grams, running on the Wear OS at 360×360 pixels in the study (Fig. 9). We decided to use a circular watch in the study since it is the most popular shape for (smart)watches [23, 27]. We developed SwipeRing with the Android Studio 3.4.2, SDK 28. We collected the original source code of C-QWERTY from Costagliola et al. [7], which was also developed for the Wear OS. Both applications calculated all performance metrics directly and logged all interactions with timestamps.

### 6.2 Design

We used a between-subjects design to avoid interference between the conditions. Since both techniques use similar layouts, the skill acquired while learning one technique would have affected performance with the other technique [33]. There were separate groups of twelve participants for C-QWERTY and SwipeRing. Each group used the respective technique to enter short English phrases in eight blocks. Each block contained 10 random phrases from a set [34]. Hence, the design was as follows.

> 2 groups: C-QWERTY and SwipeRing ×
> 12 participants ×
> 8 blocks ×
> 10 random phrases = 1,920 phrases in total.

Table 4: Demographics of the C-QWERTY study. YoE stands for years of experience.

| | |
|---|---|
| Age | 21–34 years (M = 25.8, SD = 3.92) |
| Gender | 3 female, 9 male |
| Handedness | 11 right, 1 left |
| Owner of smartwatches | 5 (M = 0.8 YoE, SD = 1.4) |
| Experienced gesture typists | 3 (M = 4.7 YoE, SD = 2.5) |

Table 5: Demographics of the SwipeRing study. YoE stands for years of experience.

| | |
|---|---|
| Age | 21–28 years (M = 24.8, SD = 2.33) |
| Gender | 4 female, 8 male |
| Handedness | 10 right, 1 ambidextrous, 1 left |
| Owner of smartwatches | 6 (M = 1.2 YoE, SD = 0.9) |
| Experienced gesture typists | 3 (M = 2.7 YoE, SD = 0.9) |

### 6.3 Participants

Twenty-four participants took part in the user study. They were divided into two groups. Table 4, 5 present the demographics of these groups. Almost all participants chose to wear the smartwatch on their left hand and perform the gestures using the index finger of the right hand (Fig. 9). All participants were proficient in the English language. In both groups, three participants identified themselves as experienced gesture typists. However, none of them used the method dominantly, instead frequently switched between tap typing and gesture typing for text entry. The remaining participants never or very rarely used gesture typing on their devices. Initially, we wanted to recruit more experienced gesture typists to compare the performance of inexperienced and experienced users to investigate whether the gesture typing skill acquired on mobile devices transferred to SwipeRing. But we were unable to recruit experienced gesture typists after months of trying. This strengthens our argument that gesture typing is still not a dominant method of text entry, regardless of being much faster than tap typing [30], and using SwipeRing may encourage some users to apply the acquired skill on mobile devices. All participants received a small compensation for participating in the study.

### 6.4 Performance Metrics

We calculated the conventional *words per minute* (WPM) [2] and *total error rate* (TER) performance metrics to measure the speed and accuracy of the keyboard, respectively. TER [51] is a commonly used error metric in text entry research that measures the ratio of the

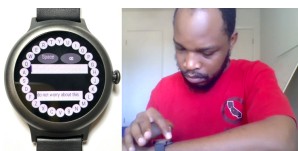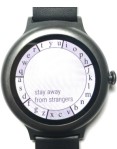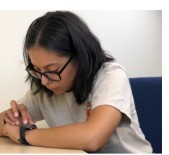

Figure 9: The device with C-QWERTY and a participant volunteering in the study over Zoom (left). The device with SwipeRing and a volunteer participating in the study (right).

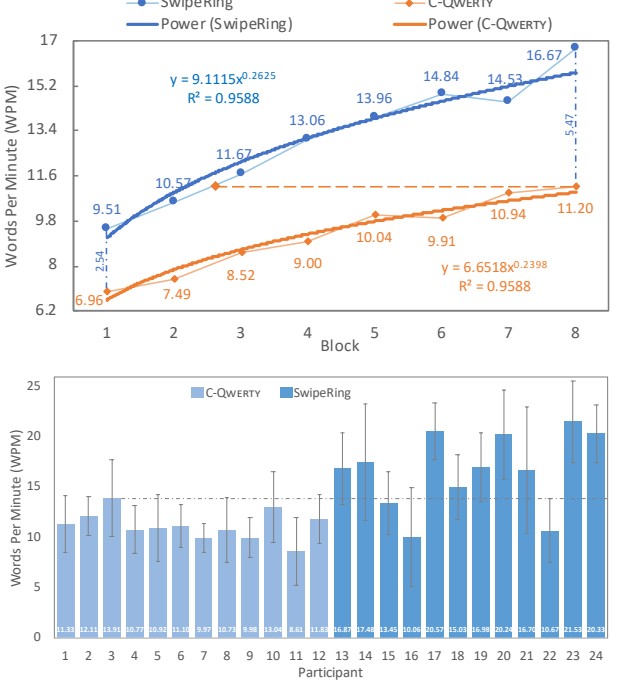

Figure 10: Average entry speed (WPM) per block fitted to a power trendline (top). The SwipeRing group surpassed the C-QWERTY group's maximum entry speed by the third block. Note the scale on the vertical axis. Average entry speed (WPM) with the two techniques for each participant in the final block (bottom).

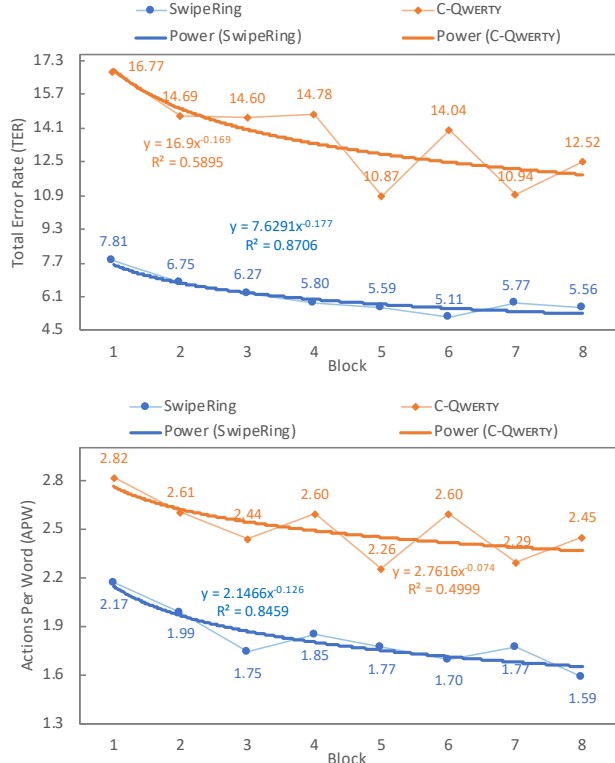

Figure 11: Average total error rate (TER) (top) and actions per word (APW) (bottom) in each block fitted to a power trendline. Note the scale on the vertical axis.

total number of *incorrect* characters and *corrected* characters to the total number of *correct*, *incorrect*, and *corrected* characters in the transcribed text. We also calculated the *actions per word* metric that signifies the average number of actions performed to enter one word. An action could be a gesture performed to enter a word, a tap on the suggestion bar, or a gesture to delete an unwanted word or letter.

### 6.5 Procedure

The study was conducted in a quiet room, one participant at a time. First, we introduced the keyboards to all participants, explained the study procedure, and collected their consents. We then asked them to complete a short demographics and mobile usage questionnaire. We instructed participants to sit on a chair, wear the smartwatch on their preferred hand, and practice with the keyboard they were assigned to by transcribing two short phrases. These practice phrases were not included in the main study. Interestingly, all participants decided to wear the smartwatch on their left hand and perform the gestures using the index finger of the other hand. The actual study started after that. There were eight blocks in each condition, with at least 5-10 minutes gap between the blocks. In each block, participants transcribed ten random short English phrases from a set [34] using either C-QWERTY_Gesture or SwipeRing. Both applications presented one phrase at a time at the bottom of the smartwatch (Fig. 9). Participants were instructed to read, understand, and memorize the phrase, transcribe it *"as fast and accurate as possible"*, then double-tap on the touchscreen to see the next phrase. The transcribed text was displayed on the top of the smartwatch. Error correction was recommended but not forced. After the study, all participants completed a short post-study questionnaire that asked them to rate various aspects of the keyboard on a 7-point Likert scale. It also enabled participants to comment and give feedback on the examined keyboards.

Due to the spread of COVID-19, the C-QWERTY group partic-

ipated in the study via Zoom, a teleconference application. We personally delivered the smartwatch to each participant's mailbox and scheduled individual online sessions with them. They were instructed to join the session from a quiet room. All forms were completed and signed electronically. Apart from that, an online session followed the same structure as a physical session. A researcher observed and recorded a complete study session. We picked up the devices after the study. The device, the charger, and the container were disinfected before delivery and after pickup.

### 6.6 Results

A Shapiro-Wilk test revealed that the response variable residuals were normally distributed. A Mauchly's test indicated that the variances of populations were equal. Hence, we used a Mixed-design ANOVA for one between-subjects and one within-subjects factors (technique and block, respectively). We used a Mann-Whitney U test to compare user ratings of various aspects of the two techniques.

#### 6.6.1 Entry Speed

An ANOVA identified a significant effect of technique on entry speed ($F_{1,22} = 25.05, p < .0001$). There was also a significant effect of block ($F_{7,22} = 63.65, p < .0001$). The technique × block interaction effect was also statistically significant ($F_{7,154} = 4.02, p < .0005$). Fig. 10 (top) illustrates average entry speed for both techniques in each block, fitted to a function to model the power law of practice [4]. In the last block, the average entry speed with C-QWERTY and SwipeRing were 11.20 WPM (SD = 3.0) and 16.67 WPM (SD = 5.36), respectively. Nine users of SwipeRing yielded a much higher entry speed than the maximum entry speed reached with C-QWERTY, illustrated in Fig. 10 (bottom). The highest average

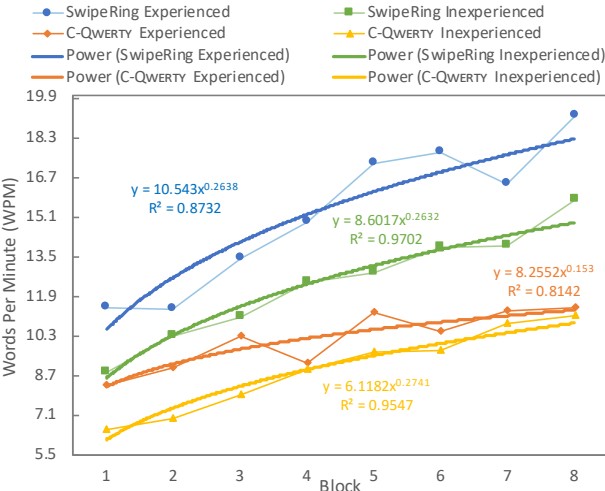

Figure 12: Average entry speed (WPM) per block for the two user groups with the two techniques fitted to power trendlines. Note the scale on the vertical axis.

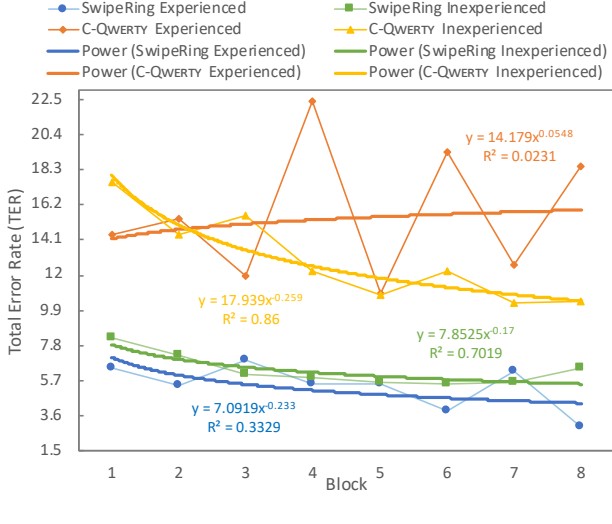

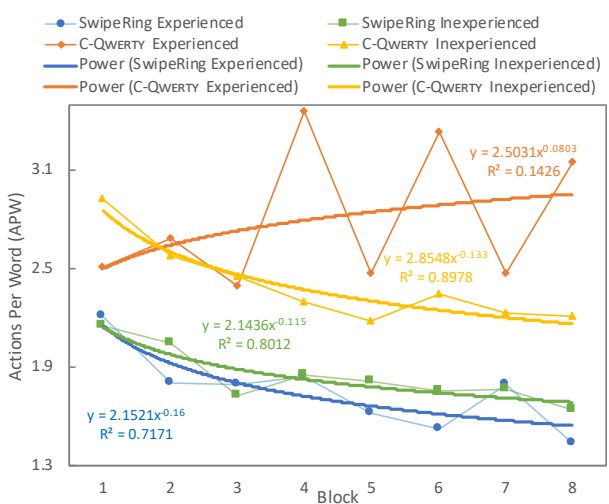

Figure 13: Average error rate (TER) (top) and average actions per word (APW) (bottom) per block for the two user groups with the two techniques fitted to power trendlines. Note the scale on the vertical axis.

entry speed in the last block was 21.53 WPM (P23, inexperienced gesture typist).

### 6.6.2 Error Rate

An ANOVA identified a significant effect of technique on error rate ($F_{1,22} = 24.61, p < .0001$). There was also a significant effect of block ($F_{7,22} = 2.89, p < .01$). However, the technique × block interaction effect was not significant ($F_{7,154} = 1.01, p > .05$). Fig. 11 (top) illustrates average error rate for both techniques in each block, fitted to a function to model the power law of practice [4]. In the last block, the average error rates with C-QWERTY and SwipeRing were 12.52% (SD = 13.91) and 5.56% (SD = 8.53), respectively.

### 6.6.3 Actions per Word

An ANOVA identified a significant effect of technique on actions per word ($F_{1,22} = 10.31, p < .005$). There was also a significant effect of block ($F_{7,22} = 3.14, p < .005$). However, the technique × block interaction effect was not significant ($F_{7,154} = 0.61, p > .05$). Fig. 11 (bottom) illustrates average actions per word for both techniques in each block, fitted to a function to model the power law of practice [4]. In the last block, the average actions per word with C-QWERTY and SwipeRing were 2.45 (SD = 1.64) and 1.59 (SD = 0.72), respectively.

### 6.7 Inexperienced vs. Experienced Gesture Typists

Although there were not enough data points to run statistical tests to compare the two user groups, average performance over blocks suggests that learning occurred with both experienced and inexperienced gesture typists with both techniques. The average entry speed over block correlated well with the power law of practice for C-QWERTY with both user groups (experienced: $R^2 = 0.8142$, inexperienced: $R^2 = 0.9547$), also for SwipeRing (experienced: $R^2 = 0.8732$, inexperienced: $R^2 = 0.9702$), illustrated in Fig. 12. The average error rate over block for both techniques, in contrast, correlated well with the power law of practice [4] for inexperienced participants (C-QWERTY: $R^2 = 0.86$, SwipeRing: $R^2 = 0.7019$), but not for experienced participants (C-QWERTY: $R^2 = 0.0231$, SwipeRing: $R^2 = 0.3329$). The average actions per word over block yielded a similar trend for C-QWERTY, where learning was observed with inexperienced participants ($R^2 = 0.8978$), but not with experienced participants ($R^2 = 0.1426$). However, with SwipeRing,

both user groups continued improving with practice (experienced: $R^2 = 0.7171$, inexperienced: $R^2 = 0.8012$), see Fig. 13.

### 6.8 User Feedback

A Mann-Whitney U test identified a significant effect of technique on willingness to use ($U = 21.0, Z = -3.1, p < .005$), perceived speed ($U = 22.5, Z = -3.07, p < .005$), and perceived accuracy ($U = 27.0, Z = -2.72, p < .01$). However, there was no significant effect on ease of use ($U = 48.0, Z = -1.5, p > .05$) or learnability ($U = 66.0, Z = -0.37, p > .05$). Fig. 14 illustrates median user ratings of all investigated aspects of the two keyboards on a 7-point Likert scale.

## 7 DISCUSSION

SwipeRing reached a competitive entry speed in only eight short blocks. It was 33% faster than C-QWERTY. The average entry speed with C-QWERTY and SwipeRing were 11.20 WPM and 16.67 WPM, respectively. Four participants reached over 20 WPM with SwipeRing (Fig. 10, bottom). Further, the SwipeRing group surpassed the C-QWERTY group's maximum entry speed by the third

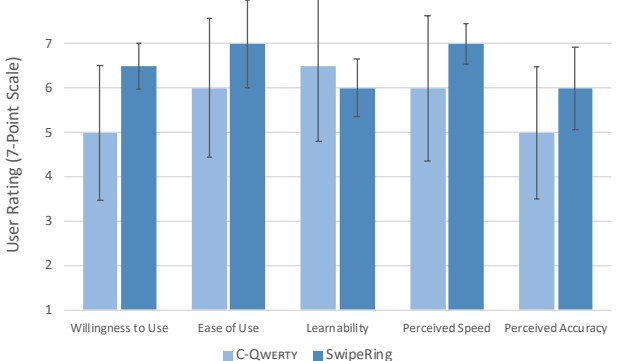

Figure 14: Median user ratings of the willingness to use, ease of use, learnability, perceived speed, and perceived accuracy of SwipeRing and C-QWERTY on a 7-point Likert scale, where "1" to "7" represented "Strongly Disagree" to "Strongly Agree". The error bars signify ±1 standard deviations (SD).

block (Fig. 10, top). It also performed better than all popular ring-shaped text entry techniques (Table 2) and some QWERTY-based techniques (Table 1) for smartwatches. Yi et al. [62] and Watch-Writer [13] reported much higher entry speed than SwipeRing. Both techniques use aggressive statistical models with a miniature QWERTY to account for frequent incorrect target selection due to the smaller key sizes (the "fat-finger problem" [55]). This makes entering out-of-vocabulary words difficult with these techniques. In fact, the former technique does not include a mechanism for entering out-of-vocabulary words [62, p. 58]. DualKey [16] and SwipeBoard [5] also reported higher entry speed than SwipeRing. However, DualKey depends on external hardware to distinguish between different fingers and has a steep learning curve (the reported entry speed was achieved in the 15th session). SwipeBoard, on the other hand, was evaluated on a tablet computer, hence unclear whether the reported entry speed can be maintained on an actual smartwatch. Besides, all of these keyboards occupy about 45–85% of the screen real-estate, leaving a little room for displaying the entered text, let alone additional information. There was a significant effect of block and technique × block on entry speed. Entry speed increased by 38% with C-QWERTY and 43% with and SwipeRing in the last block compared to the first. The average entry speed over block for both techniques correlated well with the power law of practice [4] ($R^2 = 0.9588$). However, the learning curve for C-QWERTY was flattening out by the last block, while SwipeRing was going strong. An analysis revealed that entry speed improved by 2% with C-QWERTY and 13% with SwipeRing in the last block compared to the second-last. This suggests that SwipeRing did not reach its highest possible speed in the study. Relevantly, the highest entry speed recorded in the study was 33.18 WPM (P23, Block 6).

There was a significant effect of technique on error rate. SwipeRing was significantly more accurate than C-QWERTY (Fig. 11, top). The average error rate with C-QWERTY and SwipeRing were 12.52% and 5.56%, respectively, in the last block (56% fewer errors with SwipeRing). This is unsurprising since the designers of C-QWERTY also reported a high error rate with the technique (20.6%) using the same TER metric [7] that accounts for both corrected and uncorrected errors in the transcribed text [51]. Most text entry techniques for smartwatches report character error rate (CER) that only accounts for the uncorrected errors in the transcribed text [2]. Most errors with C-QWERTY were committed due to incorrect target selection since the keys were too small. SwipeRing yielded a lower error rate due to the larger zones that

were designed to accommodate precise target selection. There was a significant effect of block on error rate. Participants committed 13% fewer errors with C-QWERTY and 29% fewer errors with SwipeRing in the last block compared to the first. The average error rate over block correlated moderately for C-QWERTY ($R^2 = 0.5895$) but well for SwipeRing ($R^2 = 0.8706$) with the power law of practice [4]. Hence, it is likely that SwipeRing will become much more accurate with practice. Actions per word yielded a similar pattern as error rate. SwipeRing consistently required fewer actions to enter words than C-QWERTY (Fig. 11, bottom). C-QWERTY and SwipeRing required on average 2.45 and 1.59 actions per word in the last block, respectively (35% fewer actions with SwipeRing). This is mainly because participants performed fewer corrective actions with SwipeRing than C-QWERTY. There was also a significant effect of block. The average actions per word over block correlated well for SwipeRing ($R^2 = 0.8459$) but not for C-QWERTY ($R^2 = 0.4999$) with the power law of practice [4]. This suggests that actions per word with SwipeRing is likely to further improve with practice.

Qualitative results revealed that the SwipeRing group found the examined technique faster and more accurate than the C-QWERTY group (Fig. 14). These differences were statistically significant. Consequently, the SwipeRing group was significantly more interested in using the technique on their devices than the C-QWERTY group. However, both techniques were rated comparably on ease of use and learnability, which is unsurprising since both techniques used similar layouts.

We compared the performance of C-QWERTY in our study with the results from the literature to find out whether conducting the study remotely affected its performance. Costagliola et al. [7] reported a 7.7 WPM entry speed with a 20.6% error rate on a slightly larger smartwatch using the same phrase set in a single block containing 6 phrases. In our study, C-QWERTY yielded a comparable 7 WPM and 16.8% error rate in the first block containing 10 phrases.

### 7.1 Skill Transfer from Virtual QWERTY

Although there were not enough data points to run statistical tests, average performance over blocks suggests that experienced participants were performing much better with both techniques from the start. Fig. 12 shows that experienced participants consistently performed better than inexperienced participants. This suggests that experienced participants were able to transfer their smartphone gesture typing skills to both techniques. However, with C-QWERTY, inexperienced participants almost caught up with the experienced participants by the last block. While with SwipeRing, both user groups were learning at comparable rates in all blocks. Besides, both experienced and inexperienced participants constantly performed better with SwipeRing than C-QWERTY. These indicate towards the possibility that optimizing the zones for gesture similarities facilitated a higher rate of skill transfer. As blocks progressed, experienced participants were most probably more confident, consciously or subconsciously, in applying their gesture typing skills to SwipeRing.

Interestingly, error rate and actions per word patterns were quite different from the patterns observed in entry speed. With SwipeRing, experienced participants were consistently better than inexperienced users, while inexperienced participants were learning to be more accurate. We speculate this is because experienced participants made fewer errors than inexperienced participants, which required performing fewer corrective actions (a phenomenon reported in the literature [3]). In contrast, with C-QWERTY, experienced participants committed more errors, requiring more corrective actions. In Fig. 13, one can see that experienced participants' error rates and actions per word went up and down in alternating blocks. We do not have a definite explanation for this, but based on user comments we speculate that this is because experienced participants were trying to apply their gesture typing skills to C-QWERTY, only committing

more errors due to the smaller target size, then reduced speed in the following block to increase accuracy (i.e., the speed-accuracy trade-off). This process continued till the end of the study. Interestingly, with SwipeRing, both inexperienced and experienced gesture typists were improving their average actions per word with practice. It could be because participants gradually learned how to exploit the wider zones of the layout to reduce the number of incorrect actions. This suggests that optimizing the zones for precise target selection facilitated a higher rate of skill acquisition.

## 8 CONCLUSION

We presented SwipeRing, a ring-shaped keyboard arranged around the bezel of a smartwatch to enable gesture typing with the support of a statistical decoder. It also enables character-based text entry by using a multi-tap like approach. It divides the layout into seven zones and maintains a resemblance to the standard QWERTY layout. Unlike most existing solutions, it does not occupy most of the touchscreen real-estate or require repeated actions to enter most words. Yet, it employs the whole screen for drawing gestures, which is more comfortable than drawing gestures on a miniature QWERTY. The keyboard is optimized for target selection and to maintain similarities between the gestures drawn on a smartwatch and a virtual QWERTY to facilitate skill transfer between devices. We compared the technique with C-QWERTY, a similar technique that uses almost the same layout to enable gesture typing, but does not divide the keyboard into zones or optimize the zones for target selection and gesture similarity. In the study, SwipeRing yielded a 33% higher entry speed, 56% lower error rate, and 35% lower actions per word than C-QWERTY in the last block. The average entry speed with SwipeRing was 16.67 WPM, faster than all popular ring-shaped and most QWERTY-based text entry techniques for smartwatches. Results indicate towards the possibility that skilled gesture typists were able to transfer their skills to SwipeRing. Besides, participants found the keyboard easy to learn, easy to use, fast, and accurate, thus wanted to continue using it on smartwatches.

## 9 FUTURE WORK

There are multiple possible extensions to our work. First, the proposed keyboard could be useful in saving touchscreen real-estate on larger devices, such as smartphones and tablets. The keyboard could appear on the screen like a floating widget, where users perform gestures to enter text. Second, the SwipeRing can be used in virtual and augmented reality by using a smartwatch or different types of controllers. Finally, there is a possibility of an eyes-free text entry with SwipeRing. We speculate, when the positions of the zones are learned, users would be able to perform the gestures without visual aid. This can make the whole touchscreen available to display additional information by making the keyboard invisible.

### 9.1 Limitations

We discussed several limitations of the work. To summarize: first, we had different numbers of experienced and inexperienced participants in the study. The sample size was also small. Hence, a definitive conclusion cannot be drawn about the transference of skill between devices. Second, due to the spread of COVID-19, we had to switch to an online format mid-study, as in-person studies were unsafe. This resulted in one condition being studied in-person, while the other online. However, based on the performance reported in prior work, we speculate that this did not impact the findings of this research. Finally, we did not investigate the proposed method in mobile settings, such as while walking or commuting. However, we anticipate our method to perform much better in such scenarios compared to the conventional methods since it does not always require precise target selection.

## ACKNOWLEDGMENTS

We thank Costagliola, Rosa, D'Arco, Gregorio, Fuccella, and Lupo for providing us with the original source code of C-QWERTY [7]. This project was supported in part by a Senate Faculty Award and a Graduate Student Opportunity Program (GSOP) Fellowship at UC Merced.

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
