# OpenReview forum: "SwipeRing: Gesture Typing on Smartwatches Using a Segmented Qwerty Around the Bezel"
_graphicsinterface.org/Graphics_Interface/2021/Conference — GI 2021_

### Official Review · AnonReviewer1 · 2021-01-12
**SwipeRing: Gesture Typing on Smartwatches Using a CircularSegmented QWERTY**

**Rating:** 9
**Confidence:** 4

**Review:**


The authors present a new type of gesture-based binned-keys circular keyboard for smart watch text entry. They descibe their design process, how they decided on layout, their motivation, and an experimetn they conducted comparing their method to a comparable, roughly competitive other keyboard. They find that their version can produce higher WPM and lower error rates than other comparable keyboards, and compares favorably to other types of keyboards for smartwatches.

Pros:
- excellently written, clear
- good motivation, clear method (reproducible)
- use of standard metrics helps compare to existing studies
- great use of figures and graphs
- good statistical tests, and relys on good descriptive statistics more

cons:
- seems to argue strongly for negatives of other approaches that seem a bit off
- bare minimum participant pool for between subjects
- strong learnability motivation but due to covid, participant sampling, couldn't evaluate this aspect strongly (postive: they did try)

Overall I enjoyed this paper. It was well written and easy to understand. Even the more detailed math/simulation sections where they designed the smartwatch layout was clearer than most mathy sections of other papers.  Thus, the contribution should be clear to readers of the paper

The only writing issue I found was a strong repeated argument on how some other top-performing methods rely on aggressive statistical modeling and how this precludes (or makes difficult) non-dictionary vocabulary. I would argue this is false (while I am less familiar with the other techniques). This is not a fault of aggressive statistical prediction (the authors themselves leverage prediction in the presented technique), but of enabling such in the interface. For example, dwelling and other gestures suggested for this technique to add features could also be used to add precise but slow text entry to other techniques. So I think such strong claims should be reduced. As it stands, this technique uses less screen real estate (and perhaps a better layout) and still performs well so I don't think the authors need to be so defensive. Otherwise, the motivations were very clear and fed well into the design of the interface.

In terms of the experiment, I found the method clear and would be easy to reproduce. The chocie of measures was good, and enabled the authors to compare their implementation of other techniques to the existing results in the literature, adding replication data and more validity to their study. While I thought the number of participants was low for a between-participants design, without a pre-experiment power analysis the experiment does pass a bare-minimum rule of thumb number of participants. In times where running experiments is difficult, I find this quite acceptable.

On that note, with such a focus on skill transfer, it was really unfortunate that a experienced-person analysis could not be carried out (statistically). I do think the authors made an interesting case with great descriptive statistics with the few participants they had. Further, on this note, I think the use of figures and presentation of descriptive statistics was excellent, clear, and very informative. This gave the work much clearer results and generalizability rather than relying on just base p values and effect sizes alone.

So in summary, there were some small drawbacks but I find this paper to use strong methods, has interesting results that stem from a well designed study anchored in clear motivation. I would recommend this for acceptance.



Small points:
I would consider reorganizing 7.1 into a part of the analysis and results and then into the discussion. It felt strange to introduce new data and analysis (the correlations, etc) so late in the paper. Not required, just a suggestion.

references justifying why a score of ~115 on Procrustes analysis is good should be provided. E.g., later you also say 6-19 is good. Not everyone reading this paper will have worked with swipe gesture and common analysis again so a few sentences on it would really help expand the paper's audience.

6.5 last paragraph,. second last sentence: piked  should be "picked"

---

### Official Review · AnonReviewer2 · 2021-01-14
**The paper presents SwipeRing, a circular keyboard designed for smartwatch text entry. A solid and interesting paper to read. Recommended for acceptance.**

**Rating:** 9
**Confidence:** 4

**Review:**

The paper presents SwipeRing, a circular keyboard designed for smartwatch text entry that arranges QWERTY layout into several segmented zones. The experimental evaluation compared the proposed design to a more traditional circular keyboard design that doesn't arrange keys into zones, showing faster entry speed and lower error rate with the proposed design.

As an HCI researcher not working on the specific topic of text entry design and wearable interfaces, I found this paper interesting to read, and the body of research work presented in the paper well-constructed and delivered. The building and construction of the SwipeRing keyboard appear to be really solid to a general audience, which has carefully motivated and justified the proposed design along with necessary information and details for replicating the design and implementation. So overall the technical quality of the work appears to be very well.

As for the intellectual merits and originality of the proposed concepts, the proposed design leverages segmented QWERTY layout and skill transfer from past experience of using QWERTY to support typing on smartwatches. It’s also valuable to see a discussion on screen occupancy and a comparison of different keyboard designs on balancing visual occupancy and typing performance. While I’m not familiar with the literature of mobile text entry, the paper seems to bear sufficient novelty. Perhaps one limitation is that there’s not enough empirical support that skill transfer from past experience using a regular QWERTY keyboard is really happening. Also, the current evaluation compared SwipeRing to another circular keyboard design. It’s curious how it may look like if comparing the design to other types of non-circular keyboard design. I think the current evaluation is valuable for making lower-level design decisions by having an internal baseline. A next step could be looking outward to compare the design with something that’s more external, increasing the ecological validity of the evaluation.

One minor concern is that the Discussion is somehow repetitive, repeating results that have been presented earlier. The Skill Transfer subsection (7.1) may be considered more as a result rather than a part of the discussion.  Also very minor is that the result obtained from the post-experiment survey (Figure 12) was described as “qualitative results” (Figure 8 bottom), which seems strange to my eyes. The survey data are apparently quantitative so that statistical tests were possible.

---

### Official Review · AnonReviewer3 · 2021-01-14
**SwipeRing: Gesture Typing on Smartwatches Using a Circular Segmented QWERTY**

**Rating:** 9
**Confidence:** 4

**Review:**

This paper contributes the design and evaluation of SwipeRing, a novel circular keyboard designed to enable gesture typing on a smartwatch using a statistical decoder. The design uses a clever technique where the standard QWERTY keyboard is arranged into seven different zones around the edge of a smartwatch, freeing up the touchscreen real-estate (making it novel compared to other solutions in this space). A controlled user study compared SwipeRing with the C-QWERTY technique that uses a layout without keyboard zones to enable gesture typing. Key results showed that SwipeRing had 33% higher text entry speed and 56% lower error rate in comparison to C-QWERTY.

This is a largely well-written paper which tackles an interesting and important problem of improving text entry on smartwatches. The design has been clearly presented along with a thorough explanation of the layout optimization and related gesture modelling techniques.  The user study description is clear and the study can be reproduced based on the details provided.

I do have some reservations about the paper that are mostly related to the user study design. One key issue is the choice of comparing SwipeRing to C-QWERTY since C-QWERTY doesn’t appear to be the state-of-the-art in circular keyboard layouts.  I understand the author’s argument that they were trying to tease out the benefits of the zone-based layout around the smartwatch bezel, but this makes the contribution of this work to be somewhat narrow. Second, although the choice to do a between-subject study was appropriate, I am a bit uncomfortable with seeing statistical findings and claims drawn from only 12 participants in each condition. Furthermore, the two study conditions were completely different: the C-QWERTY group participated fully online vs. the SwipeRing group participated in an in-person study. I am cognizant of the difficulties of running user studies during a pandemic, but I just wanted to raise issues around potential threats to validity of the results. Perhaps the discussion section would benefit from having an explicit limitations section.

I was a bit confused with the start of the discussion section as it seemed to continue to mention new study results rather than reflect on the contributions of the work or broader implications. Some of the text on page 8 can be significantly condensed and/or integrated into the results section. Although the discussion around skill transfer was interesting, the lack of statistical findings is somewhat problematic given the motivation and claims described earlier in the paper. To the extent it is possible, I would suggest that the authors tone down skill transfer as one of their key motivation (e.g., in 2.2) and successful aspects of their design.

Text entry is a fairly crowded space in HCI research with many novel solutions have already been explored (including circular layouts). In this regard, SwipeRing appears to make an incremental contribution, yet a unique one given the use of zones to lay out the keyboard around the smartwatch bezel which clearly has some benefits. Overall, despite some weaknesses, this paper makes a reasonable contribution to input and interaction techniques in HCI and I would recommend this paper for acceptance at Graphics Interface.

---

### Meta-Review · Area_Chair1 · 2021-01-15

**Recommendation:** Accept
**Confidence:** 5

**Metareview:**

This paper received three reviews from reviewers who are all fairly confident about their assessment and rate this paper as a strong accept (within the Top 15% of accepted papers). All of the reviewers agree that the paper is well-written and presents an interesting and compelling novel design for improving text entry on smartwatches. There were some concerns raised around the ecological validity of the user study and the small number of participants in the between-subject experiment (R1, R2, R3). However, given the constraints of Covid, reviewers did not see this as a major red flag for acceptance. There are other suggestions from reviewers for reconciling the discussion section and moving some of the content to the results section (R1, R2, R3) and for addressing the other approaches in related work more fairly (R1). There are several other minor editorial suggestions which the authors should consider addressing in their final revisions.

Overall, this paper is making a clear new contribution to HCI and the consensus is to accept the paper for GI 2021.

---

### Decision · Program_Chairs · 2021-01-16

Accept